# Meta-analysis reveals negative but highly variable impacts of invasive alien species across terrestrial insect orders

Grace L. V. Skinner [1] ✉, Rob Cooke [1], Helen E. Roy [1,2], Nick J. B. Isaac [1], Charlotte L. Outhwaite[3,4], James Rodger [5,6] & Joseph Millard [7]

Insects are crucial to ecosystem functioning but face numerous threats, with invasive alien species likely among the most severe. As insect declines continue, there is a growing need to synthesise evidence on how invasive alien species affect insects, as research has historically focused more on insects as invaders than as victims. Here we conduct a global meta-analysis encompassing 318 effect sizes across 52 studies, assessing invasive alien species impact on terrestrial insect orders (Coleoptera, Hemiptera, Hymenoptera, and Orthoptera), and examining factors influencing these effects. We show that invasive alien species reduce the abundance of insects included in our study by 31%, and species richness by 26%, though these impacts are highly variable across taxa. Stronger negative impacts are found for invasive alien animals compared to invasive alien plants, and for Hemiptera (true bugs) and Hymenoptera (bees, wasps, ants) compared to Coleoptera (beetles). These findings provide quantitative estimates for the relative vulnerability of insects to invasive alien species, which is an important step towards halting declines.

Insects are one of the most abundant and species-rich groups on land[1], but are undergoing concerning declines across the world[2–8]. If this trend continues, ecosystem services such as pollination, pest control, decomposition, and food web stability[9–11] will be further threatened, leading to adverse impacts on global biodiversity and human well-being[12].

Invasive alien species are likely one of the greatest threats to insect biodiversity[8,12,13] and are being introduced worldwide in increasing numbers[14]. Alien species are those introduced outside their natural range, unintentionally or intentionally, to new areas where they would not naturally occur via human activities such as trade and tourism[15]. Once an alien species establishes and spreads it is termed an invasive alien species[14,15]. The negative impacts of invasive alien species[15–18] occur when invasive alien species predate or parasitise native species, compete for resources, transmit pathogens and diseases, or hybridise with natives[19], leading to homogenisation of biota and driving global extinctions[14,15,20]. Nevertheless, the effects of invasive alien species are not always negative[21–24]. For example, invasive alien plants can provide pollen and nectar to native pollinators[25–27], or invasive alien fish can become a food source for native predatory fish[28].

The drivers of differing responses to invasive alien species remain unclear[29,30], particularly for insects, despite the vital role insects play in ecosystems. Previous research syntheses have focused on the impact of invasive alien insects on species more widely, rather than the impact of all invasive alien species on insects specifically[31–34]. Several other meta-analyses have considered the impact of invasive alien species on animals more broadly—including, but not focusing on, insects—often highlighting negative but highly variable effects[35–37]. Syntheses have

[1]UK Centre for Ecology & Hydrology, Maclean Building, Wallingford, Oxfordshire, UK. [2]Centre for Ecology and Conservation, University of Exeter, Penryn, UK. [3]Centre for Biodiversity & Environment Research, University College London, London, UK. [4]Institute of Zoology, Zoological Society of London, Outer Circle, Regent's Park, London, UK. [5]Department of Mathematical Sciences, Stellenbosch University, Stellenbosch, South Africa. [6]Department of Biological and Agricultural Sciences, Sol Plaatje University, Kimberley, South Africa. [7]Department of Zoology, University of Cambridge, Cambridge, UK. ✉e-mail: GraSki@ceh.ac.uk

also addressed the impact of other threats on insect biodiversity including urbanisation[38], plantations[39], dams[40], and nutrient enrichment[41,42]. There is a clear need to better understand how insects specifically are affected by invasive alien species to better inform conservation action and to add to a developing evidence base on threats to insects[19,40,41,43].

Taxonomy, geography, and traits are all likely predictors of the impact of invasive alien species on insects. For taxonomy, an invasive alien animal may have a more severe and immediate impact than an invasive alien plant due to its increased potential for direct interactions with the native insect via competition and predation, and vice versa[44]. Geographical factors such as geographical realm are also likely to have a substantial impact. For example, given they are often more specialised and thus more sensitive to change, insects have been found to be more susceptible to invasive alien species inside the tropics than outside[39]. Alternatively, it may be more challenging for alien species to establish and impact native populations in tropical regions due to high levels of competition[45] or lack of disturbance[46]. Evidence also strongly suggests that insects on islands will be more negatively affected by invasive alien species due to their isolated geographical ranges and the difficulty of recolonising after extinction[15,47]. For traits, characteristics such as flight capability influence mobility of the native insect, potentially allowing the native to escape areas disturbed by invasive alien species (provided there is suitable habitat available), thereby reducing the impact of an invasion[48].

Here we present a meta-analysis of the impact of invasive alien species on a subset of terrestrial insect biodiversity. While previous research has examined the effect of invasive alien species (specifically invasive alien plants) on Lepidoptera[49], we focus on insects in the primarily terrestrial orders Coleoptera (beetles), Hemiptera (true bugs), Hymenoptera (ants, bees, sawflies, and wasps), and Orthoptera (grasshoppers, locusts, and crickets). We selected these orders because invasive alien species were identified as a major potential threat in an expert elicitation process[43]. We address two key research questions: 1. What is the impact of invasive alien species on the abundance, biomass, and species richness of insects in the taxonomic orders Coleoptera, Hemiptera, Hymenoptera, and Orthoptera, relative to areas without invasive alien species present? 2. How do our moderator variables influence the magnitude of this effect? Our expectation was that insect biodiversity will be lower in areas with invasive alien species, but that this effect will be moderated by native insect taxonomy, invasive alien taxonomy (i.e., animal or plant), geographical realm (i.e., tropical or non-tropical), island invasions (i.e., an island or continental invasion), and flight capability (i.e., flying or non-flying in the adult stage). We additionally examine the year of study publication as a potential moderator to investigate the extent to which publication date predicts the reported effect of invasive alien species[50].

## Results
### Data description
We extracted data from the 52 studies that met the predefined inclusion criteria, totalling 318 effect sizes (median effect sizes per study = 4; minimum = 1; maximum = 31) (Supplementary Fig. 1), once the single study analysing biomass was removed. Date of study publication ranged from 1995 to 2022, with more than two-thirds of the studies being published in the latter half of this range (2009 to 2022; Fig. 1a). The distribution of effect sizes shows broad spatial coverage, with data from every continent except Antarctica (Fig. 1b). Many effect sizes originate from North America ($n = 81$; 25%) and Europe ($n = 105$; 33%), reflecting wider spatial biases in insect data[51,52], while 16% of effect sizes originated from tropical biomes and 7% from islands.

Regarding the invasive alien species investigated, 30 studies assessed an invasive alien animal (including insects), while 22 assessed an invasive alien plant. Of the terrestrial insect orders investigated, most effect sizes describe the abundance or species richness of

Hymenoptera (134 effect sizes; 42%) and Coleoptera (133 effect sizes; 42%); followed by Hemiptera (43 effect sizes; 14%), and Orthoptera (8 effect sizes; 3%) (Fig. 1c). Subsequently, ants and dicotyledon plants were the most frequently reported invasive alien species in our dataset, with 39% and 34% of effect sizes, respectively (Fig. 1d). The remaining effect sizes describe the effect of other invasive alien plants and invertebrates, as well as mammals, fish, crustaceans, reptiles, and amphibians. Over 40% of the effect sizes describe how invasive alien species presence affects the focal insect taxon at the species level, and over 75% to at least the family level. The majority of effect sizes (278; 87%) describe changes in abundance, while only 40 (13%) effect sizes report changes in species richness.

### How do invasive alien species affect insect abundance and species richness across four orders?
The abundance of Hymenoptera, Coleoptera, Hemiptera, and Orthoptera was 31% lower on average (95% confidence interval: 45% to 14% lower; LRR: −0.37 [−0.60, −0.15]) when invasive alien species were present compared to absent. Moreover, species richness was 26% lower (95% confidence interval: 44% to 1% lower; LRR: −0.30 [−0.59, −0.01]) with invasive alien species (Fig. 2). Heterogeneity for abundance data, assessed with multi-level $I^2$, indicates high variation (91%), with between-study differences accounting for 28% of the variation, and within-study differences accounting for 63%. The variance among true effect sizes for abundance was partitioned into variance between studies ($\sigma^2 = 0.32$) and within studies ($\sigma^2 = 0.70$). For species richness data, heterogeneity was 98% (37% between, 60% within). The variance components for species richness were $\sigma^2 = 0.23$ between studies and $\sigma^2 = 0.37$ within studies.

For the abundance models, the funnel plots were visually symmetrical (Supplementary Fig. 2) around the overall effect size, showing no apparent publication bias. The rank correlation test (non-significant asymmetry; Kendall's tau = −0.0142, p = 0.7245) and adapted Egger's regression (no relationship between effect size and its error; estimate = 0.2463, p = 0.6112) formally supported this, indicating no concerns of publication bias. Still, the data points did not form the classic funnel shape, likely due to high heterogeneity across ecological studies, where larger studies do not necessarily show greater precision[50,53,54]. Our results did not qualitatively change under multiple sensitivity analyses, including when using Hedge's g as the effect size, when excluding influential effect sizes (Cook's distance), when excluding data collected with aquatic sampling techniques, or when excluding data where the small sample corrected standardised mean of either the treatment or control did not pass Geary's test. The AIC value was greater for a model including a phylogenetic correlation matrix as a random effect (Supplementary Table 1).

The rank correlation test for the species richness model indicated potential funnel plot asymmetry (Kendall's tau = −0.2872, $p = 0.0088$), although this was not supported by the adapted Egger's regression (estimate = 0.7596, $p = 0.3140$), which found no relationship between effect size and its error. For the sensitivity analyses, the results were less consistent: the negative effect of invasive alien species on insect species richness remained significant when excluding data collected with aquatic sampling techniques, but not when using Hedge's g as the effect size, when excluding particularly influential data points, nor when excluding data that did not pass Geary's test (Supplementary Table 1). Thus, there is some evidence of publication bias for our species richness models and they are less robust to changes in metric and data inclusion than the abundance models.

### How do the moderator variables influence the magnitude of the effect of invasive alien species?
The magnitude of the effect of invasive alien species was affected by the focal insect order (Fig. 3). Hemipteran abundance was 58% lower (72% to 37% lower) in sites where invasive alien species were present,

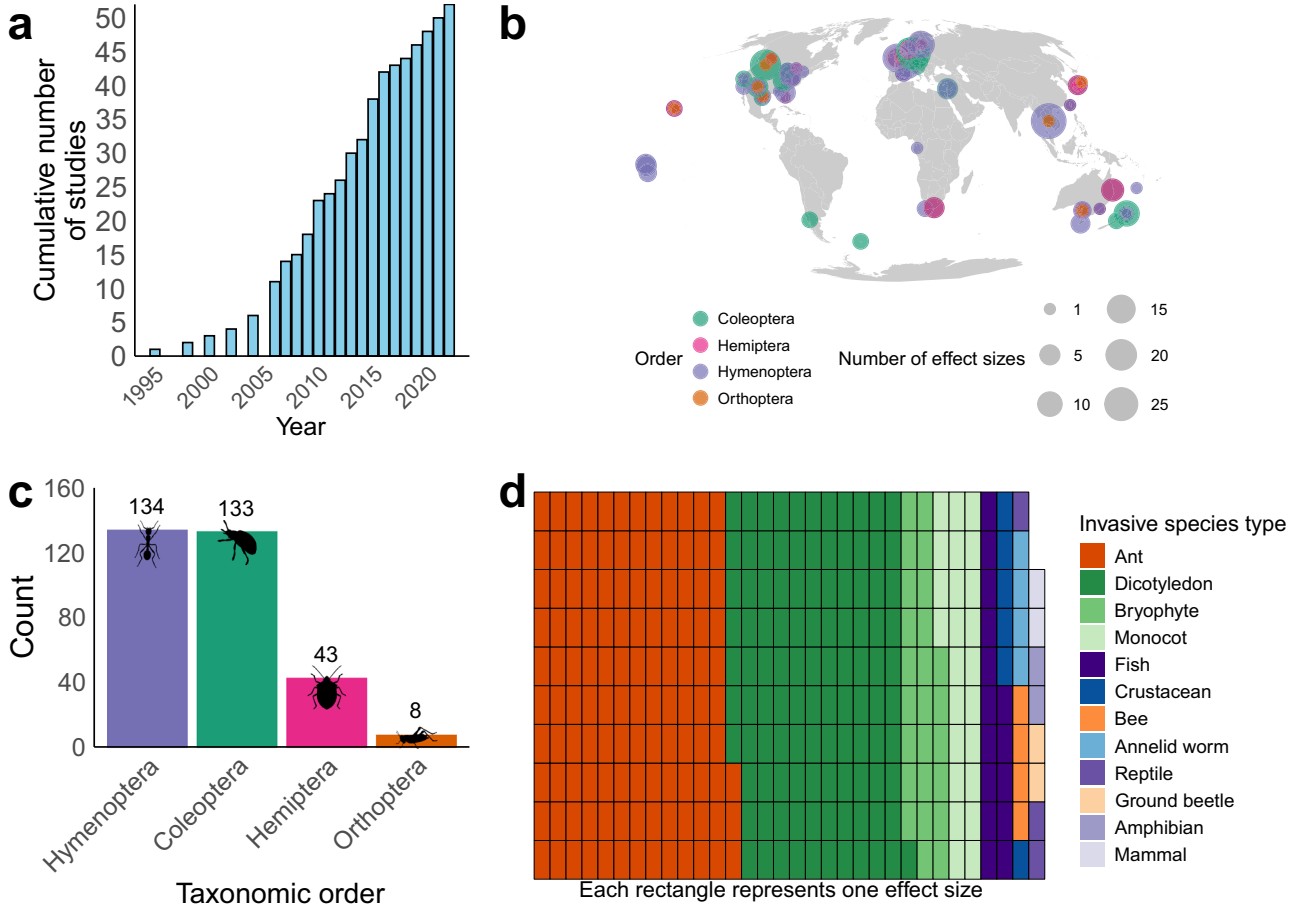

**Fig. 1 | Temporal, spatial, and taxonomic coverage of collated data.** Frequency of 318 insect biodiversity effect sizes collected according to time, geography, taxonomy of focal insect taxa, and taxonomy of invasive alien species. **a** The temporal distribution of effect sizes, showing the cumulative number of effect sizes from 1995 to 2022; **b** the global distribution of effect sizes. Colour indicates the taxonomic order of the focal taxa: Coleoptera (green), Hemiptera (pink), Hymenoptera (purple), Orthoptera (orange); size indicates number of effect sizes. Landmass polygons from the rnaturalearth[108] R package, displayed using a Mollweide projection; **c** effect sizes split by the taxonomic distribution of the focal insect taxa. The icons are all from PhyloPic.org under Public Domain Dedication 1.0 licences (see collection https://www.phylopic.org/collections/1ff41ccd-9f6b-0e7d-1197-7c73c94d7628). Creator credits: Birgit Lang, Brockhaus and Efron Encyclopedic Dictionary, Darrin Schultz, and Michael Day; **d** waffle plot showing effect sizes split by the taxonomic distribution of the invasive alien species. Each rectangle represents one effect size. The taxonomic categories are insects (orange), plants (green), vertebrates (purple), non-insect invertebrates (blue).

with Hymenopteran abundance also found to significantly decrease (−37% [−54%, −14%]). Contrastingly, the results for Coleoptera (−12% [−34%, 18%]) and Orthoptera (−27% [−68%, 66%]) were not significant (Fig. 3a). Hymenopteran species richness was 46% lower (62% to 21% lower) in the presence of invasive alien species, while no significant change was detected for Hemiptera or Coleoptera (Fig. 3b). No effect sizes were collected for Orthopteran species richness. Including focal insect order as a moderator variable in the model explained a significant proportion of the heterogeneity in both the abundance (QM = 12.7882, p = 0.0051) and species richness models (QM = 6.7964, p = 0.0334), indicating strong differences in response to invasive alien species by different insect orders. The variance among true effect sizes for abundance was partitioned into components between studies (σ² = 0.33) and within studies (σ² = 0.67). For species richness, the corresponding values were σ² = 0.19 and σ² = 0.34. Residual heterogeneity was significant for both abundance (QE = 2008.72, df = 273, p < 0.0001) and species richness (QE = 2368.89, df = 37, p < 0.0001), indicating that unaccounted variation remains in both models.

The type of invasive alien species, whether animal or plant, also moderated the overall effect on abundance (QM = 4.0595, p = 0.0439) (Fig. 4). The abundance of focal insect taxa decreased in the presence of invasive alien animals (−43% [−57%, −24%]), while no significant effect was observed for invasive alien plants (−11% [−36%, 24%]). For

species richness, the overall effect size was greater in the presence of invasive animals compared to plants, but the groups did not significantly differ (QM = 0.9556, p = 0.3283). Most of the effect sizes for invasive animals described the effect of invasive alien insects, particularly ants (123 effect sizes, 76%), while the invasive plant group was dominated by dicotyledon plants (108 effect sizes, 69%) (Fig. 1d). For abundance, variance among true effect sizes was partitioned into components between studies (σ² = 0.26) and within studies (σ² = 0.71). The corresponding values for species richness were σ² = 0.22 and σ² = 0.38. Significant residual heterogeneity remained in both models (abundance: QE = 2153.6, df = 275, p < 0.0001; species richness: QE = 3018.45, df = 38, p < 0.0001).

For abundance, single moderator models showed that tropical versus non-tropical areas, islands smaller than 25,000 km² versus mainlands, focal insect flight ability, and year of study publication did not significantly affect the results (Supplementary Fig. 3). Levels within each moderator did not differ from one another. These moderators were not assessed in relation to species richness due to the limited data available for this metric. While a multi-moderator model produced some differing results, due to reduced sample size and variance inflation factors (VIFs) indicating multicollinearity among some moderators, we had less confidence in those estimates (Supplementary Table 2).

## a (Abundance)

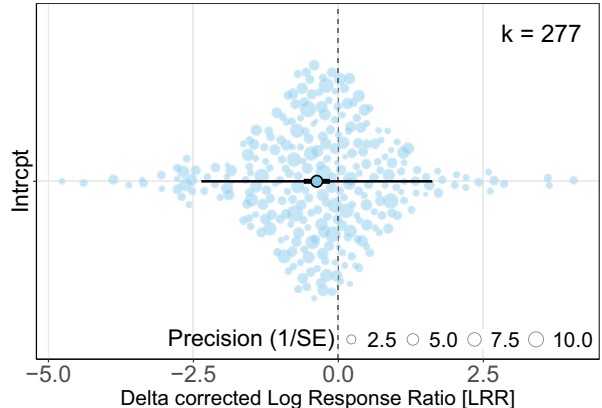

## b (Species richness)

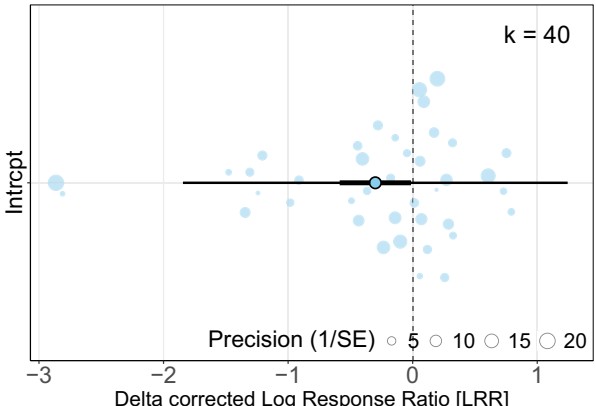

**Fig. 2 | Effect of invasive alien species on insects.** Overall model-derived effect of invasive alien species presence on the (**a**) abundance and (**b**) species richness of insects in the orders Hymenoptera, Coleoptera, Hemiptera, and Orthoptera. Plots derived from results of metafor[94] models run with (**a**) all abundance data (k = 277 effect sizes), and (**b**) all species richness data (k = 40 effect sizes), with no moderator variables. The solid dot represents the overall model-derived estimated effect size (delta corrected Log Response Ratio), with thick black bars indicating the 95% confidence intervals, and thinner black bars the prediction intervals. Effect sizes for each data point are represented by the circles, with circle size representing its weighting in the model (precision; 1 / standard error). The positioning of each circle on the y axis is so all points can be seen (jittered).

## Discussion

Here we show that, for the subset of terrestrial insect orders included in our study (Hymenoptera, Coleoptera, Orthoptera, and Hemiptera), invasive alien species reduce abundance by 31% and species richness by 26%. However, the results are highly variable and context-dependent, consistent with previous meta-analyses[35–37]. Although tests indicate some publication bias in the species richness dataset and sensitivity of estimates to data inclusion, losses of species richness exceeding 20%, as observed here, are likely to substantially impair the contribution of biodiversity to ecosystem function and services, and thus adversely affect human well-being[55]. We note that broader inclusion of terrestrial insect orders beyond those identified as having invasive alien species ranked among their top threats might reveal a more variable and on average less negative response. The most substantive impacts of invasive alien species across these insect groups include a 58% reduction in abundance for Hemiptera, and a 37% reduction in abundance and 46% reduction in species richness for Hymenoptera. The magnitude of these losses due to invasive alien species are comparable to estimates of the impacts of historical

climate warming and intensive agricultural land use on insects, where reductions of almost 50% in abundance and 27% in species richness have been estimated, relative to those in less-disturbed habitats with lower rates of historical climate warming[56].

The impacts of invasive alien species on terrestrial insects have the potential to disrupt and destabilise ecosystems[55,57,58], potentially leading to cascading effects that could alter essential insect-driven services including pollination, pest control, decomposition, and food web stability[9–12]. Any ecosystem changes due to the invasion-driven loss of insects could have knock-on effects on crop yields and food production[9,59], with consequences for human health. While the extent to which these declines translate into shifts or losses of ecosystem function has yet to be assessed[8], the threats posed by invasive alien species are expected to continue rising[14,15]. Every year, ~200 new alien species are introduced globally through human activities[14]. Moreover, the impacts of invasive alien species are predicted to be exacerbated by climate change, as climatic conditions become more favourable for the establishment of some invasive alien species and ecosystems become less resistant to biological invasions[14]. Thus, the impacts we have quantified could intensify, further affecting insect populations across the globe.

We find that invasive alien animals have stronger negative impacts on terrestrial insect abundance and species richness than invasive alien plants, in line with findings by Montero-Castaño and Vilà[36], who reported a similar trend for native pollinators. These greater impacts may be due to more direct competition between native insects and invasive alien animals for similar resources, compared to the more indirect effects of invasive alien plants, leading to more immediate effects[60]. Our findings are consistent with Tercel et al.[34], who focused on the impact of non-native ants, while we considered invasive aliens of any species. As a result, only eight of the 52 studies we identified were also present in their study. Nevertheless, the overall conclusion regarding the negative impact of invasive alien species, particularly invasive alien animals, such as ants, on insects remains consistent between studies.

A number of studies report an increase in abundance of some insects associated with invasive alien plants. For example, Lopezaraiza-Mikel et al.[26] found that plots with the invasive alien Himalayan balsam (*Impatiens glandulifera*) attracted more insect pollinators than plots without Himalyan balsam, showing how invasive alien species can cause an increase in the abundance of certain species. Similarly, Hansen et al.[61] observed that sites invaded by spotted knapweed (*Centaurea stoebe*) had higher abundance of ground beetles (Coleoptera: Carabidae), including the omnivorous *Amara* and *Harpalus*, and the carnivorous *Calosoma*, likely due to knapweed increasing direct food resources, and supporting greater prey abundance, respectively. Although we found invasive alien animals to be generally more detrimental, Freeland-Riggert et al.[62] found that riffle beetles (*Stenelmis* spp.; Coleoptera: Elmidae) benefitted from the presence of an invasive alien crayfish, likely because their unpalatability led crayfish to preferentially consume other prey, allowing *Stenelmis* spp. to thrive. Together, these examples demonstrate that while invasive alien animals often have stronger negative impacts on native insects than invasive alien plants on average, there are noteworthy exceptions. Nevertheless, assessments of the positive impacts of invasive alien species should not be used to balance or offset their negative impacts[63]. Indeed, the outcomes of biological invasions are highly context-dependent[64]. In novel ecosystems, where native vegetation has been lost, alien plants might restore some ecosystem functions whereas in natural ecosystems, invasive alien plants might out-compete and replace native species and diminish faunal communities[65]. Furthermore, ecological cascades and feedback drive community-level processes, including disruption of mutualistic interactions, and further influence the adverse outcomes of biological invasions on ecosystem

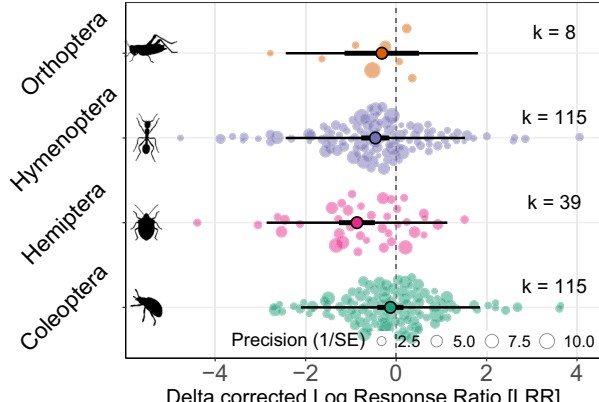

a (Abundance)

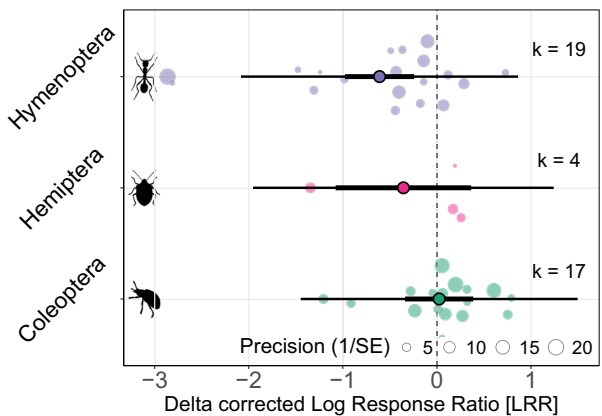

b (Species richness)

**Fig. 3 | Effect of invasive alien species on insects, split by insect order.** Model-derived response (delta corrected Log Response Ratio [LRR]) of insects to invasive alien species presence, split by taxonomic order of the focal taxa: Coleoptera (green), Hymenoptera (purple), Hemiptera (pink), Orthoptera (orange). Plots derived from results of metafor[94] models run with (**a**) all abundance data (k = 277 effect sizes), and (**b**) all species richness data (k = 40 effect sizes), with insect order as a moderator variable. k represents the number of effect sizes for each order, as indicated on the plot. Plot elements (e.g., dots, bars, circles) as in Fig. 2. The icons are all from PhyloPic.org under Public Domain Dedication 1.0 licences (see collection https://www.phylopic.org/collections/1ff41ccd-9f6b-0e7d-1197-7c73c94d7628). Creator credits: Birgit Lang, Brockhaus and Efron Encyclopedic Dictionary, Darrin Schultz, and Michael Day.

function, highlighting the complexity and challenges of predicting the impacts of invasive alien species[66].

Hemiptera and Hymenoptera were both more negatively affected than Coleoptera by invasive alien species. Given that a large number of Hemipterans feed on plants, invasive alien plants could disrupt these feeding relationships by outcompeting native plants[67]. For example, invasive alien plants such as beach rose (*Rosa rugosa*)[68], Himalayan balsam[69], and West Indian marsh grass (*Hymenachne amplexicaulis*)[70] had some of the strongest reported negative effects on Hemiptera. Notably, while West Indian marsh grass negatively affected Hemiptera, it appears to create a more favourable habitat for Coleoptera[70]. Interestingly, Tercel et al.[34] found that Hemipteran insects were the only group to increase in abundance in response to invasive alien ants, potentially because ants protect aphids for the harvest of honeydew[71]. This inconsistency in findings may be due to only 18% of the Hemipteran abundance effect sizes in our dataset involving an invasive alien ant, and only a few effect sizes where the Hemipteran was an aphid. Thus, our broader scope may have revealed wider negative impacts of

invasive alien species on Hemiptera. For Hymenoptera, their strong negative response could be explained by the large proportion of effect sizes describing a native ant in competition with an invasive alien ant, such as fire (*Solenopsis* spp.), Argentine (*Linepithema humile*), or yellow crazy ants (*Anoplolepis gracilipes*). Invasive alien ants have also been shown to impact native bees, such as those in the *Hylaeus* genus, through predation on larvae or interference with nectar feeding[72]. It is important to note that there is considerable diversity within the focal insect orders, including in life-history traits. While we show Hemiptera and Hymenoptera to be more strongly affected than Coleoptera overall, not all species within these orders will respond in the same way due to differences in factors such as feeding and social behaviour, size, and flight capability.

We found limited evidence that variation in effect sizes was explained by island invasion, geographical realm, flight capability, or year of study publication, with no significant differences in native insect abundance responses to invasive alien species between levels of these moderators. It is surprising that neither our findings nor those of Cameron et al.[31] show stronger effects on islands, given the widespread expectation that species on islands will be more severely impacted compared to those on mainlands[47,73–75]. However, with only 7% of the data in our study originating from islands, and Cameron et al.[31] noting the scarcity of studies on islands, island-specific impacts should be revisited once more data become available. Similarly, for geographical realm, only 16% of data originated from tropical countries, which likely limited our ability to draw conclusions on this variable. With a more even split between data from tropical and non-tropical zones, the results could provide evidence for whether tropical regions are more affected by invasive alien species—due to greater specialisation and sensitivity to change[39]—or less affected, as high competition and reduced disturbance can make it harder for invasive alien species to establish[45,46]. Notably, the IPBES invasive alien species assessment identifies invertebrates as a critical data gap, underscoring the urgent need to mobilise data and knowledge on insects globally to address these research deficiencies[14]. For flight capability, we could only assess this trait when the focal insect taxa were reported at a higher taxonomic resolution than order (as flight capability varies within orders), reducing the sample size for this analysis. We also used the ability to fly as a binary proxy for mobility, though defining flight ability is not always straightforward. For example, while ants are generally considered non-flyers, queens and males do fly at certain times. Finally, we did not detect temporal bias, indicating that our study did not suffer from earlier studies reporting stronger effects than more recent studies.

As expected, while the scope of the search was global, the data we compiled were spatially biased towards Europe and North America, reflecting known biases in biodiversity studies that are often exacerbated for insects[51,52,76]. Some moderator variables could not be investigated due to insufficient reporting in the primary studies. For example, invasion intensity or time since initial invasion could have helped identify potential thresholds for significant impacts on native insect biodiversity. Few studies provide this information, even though it could have a considerable impact on invasion outcomes. For example, the emerald ash borer (*Agrilus planipennis*) depletes a tree's resources over several years before moving to a new tree[77]. After the invading insect has moved on, the invasion intensity appears low again, yet the ecosystem has fundamentally changed, and the full consequences may still emerge. Understanding the temporal dimension of invasion impact on insects is a clear research gap.

Several key areas should be considered for future work. First, investigating whether invasive alien species have greater effects on specialists compared to generalists would be valuable, as generalists may be more adaptable. Second, there is potential to summarise the impacts of invasive alien species across multiple metrics of biodiversity. Although we searched for studies focused on abundance,

## a (Abundance)

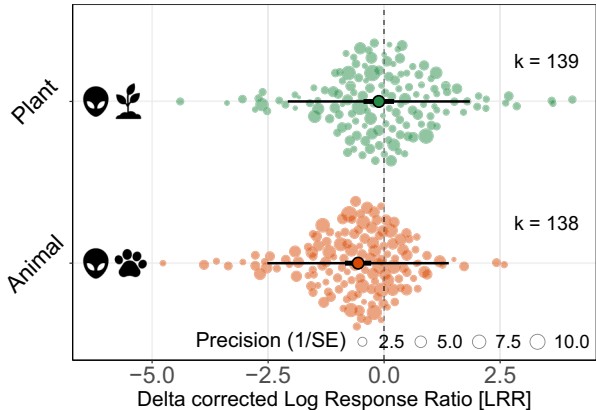

## b (Species richness)

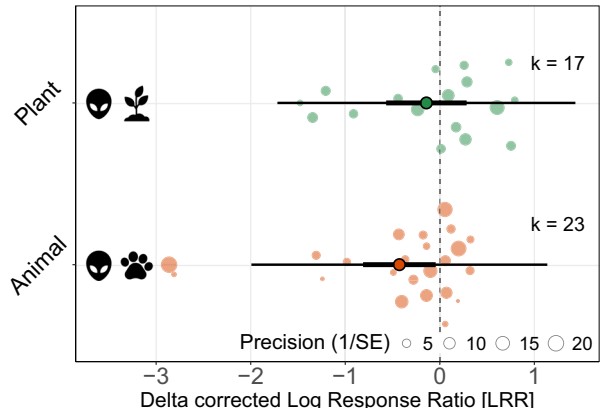

**Fig. 4 | Effect of invasive alien species on insects, split by type of invasive.**
Model-derived response (delta corrected Log Response Ratio [LRR]) of insects to invasive alien species presence, split by taxonomic order of the invasive alien species: plant (green) or animal (orange). Plots derived from results of metafor[94] models run with (**a**) all abundance data (k = 277 effect sizes), and (**b**) all species richness data (k = 40 effect sizes), with type of invasive as a moderator variable. k represents the number of effect sizes for each invasive alien type, as indicated on the plot. Plot elements (e.g., dots, bars, circles) as in Fig. 2. The icons are all from Flaticon.com under the Flaticon License (free for personal and commercial use with attribution, see collection https://www.flaticon.com/collections/NjQ0MjE2MzU=?k=1763484223725). Creator credits: Freepik, Bharat Icons, and Assia Benkerroum.

species richness, and biomass, most studies quantified abundance only. Metrics such as incidence (presence/absence), species evenness, functional diversity, and survival rate could provide more insights into the effect of invasive alien species. The lack of studies reporting incidence (presence/absence) is likely due to the exclusion of the keywords incidence and distribution in our search. Third, it is common for an ecosystem to be impacted by multiple invasive alien species simultaneously[78,79]. While we only included studies that focused on the effect of a single invasive alien species, it is possible that other undocumented invasive alien species could have been present. According to the invasion pressure effect, the negative effects are amplified with increasing numbers of introductions[78,79]. However, understanding of how the cumulative effects of multiple invasive alien species on insects develop is lacking. Lastly, similar logic can be applied to different threats. It is uncommon for threats to act in isolation[80–82], making it difficult to guarantee that observed changes are solely attributable to the invasive alien species over other threats such as land-use change. We made efforts to exclude data points where the impact of the invasive alien species was not the only threat being assessed, to avoid

confounding effects. However, more work to unpick how multiple threats interact, potentially synergistically, with invasive alien species is key to effective threat mitigation and should be prioritised[14,15,80–82].

Here we provide clear evidence that invasive alien species have overall negative, yet highly variable, effects on the abundance and species richness of terrestrial insects included in our study. Insect biodiversity is essential for many ecosystem functions and services; hence retaining these functions across landscapes will benefit both people and nature. We suggest that addressing insect declines will only be possible through dedicated commitment to understand, prevent, and manage biological invasions, and the interactions of invasive alien species with other drivers of biodiversity loss[15,83]. With limited funding available for insect conservation[84–86], increased understanding of the contexts in which insects are most affected by invasive alien species will be key for prioritising resources to ultimately inform conservation action.

## Methods

### Literature search
Following PRISMA guidelines[87,88], we collated studies assessing the impact of invasive alien species on the abundance, biomass, and species richness of our focal taxa (i.e., insects in the orders Hymenoptera, Coleoptera, Orthoptera, and Hemiptera), relative to areas without invasive alien species present (Supplementary Fig. 4). We focused on primarily terrestrial insect orders for which invasive alien species had previously been identified by experts as a major potential threat[43]. This assessment evaluated 12 insect orders (Lepidoptera, Hymenoptera, Coleoptera, Diptera, Phasmatodea, Orthoptera, Hemiptera, Dermaptera, Odonata, Ephemoptera, Plecoptera, and Trichoptera), representing 96% of described insect species. Of these, invasive alien species were ranked among the top 10 threats for Hymenoptera, Coleoptera, Orthoptera, Hemiptera, Odonata, Ephemoptera, Plecoptera, and Trichoptera. For this meta-analysis, we focused on the four primarily terrestrial orders from this group: Hymenoptera, Coleoptera, Orthoptera, and Hemiptera.

Our final search was conducted on 3rd March 2023, using both Scopus and Web of Science databases to return peer-reviewed, primary research studies. We used the following search terms: (hymenoptera OR coleoptera OR orthoptera OR grasshopper OR hemiptera) AND (invasi* OR alien OR "non native" OR introduced OR exotic OR novel) AND (abundance OR biomass OR "species richness" OR biodiversity) AND (impact OR effect OR compet*) AND NOT (distribution OR monitor* OR detect* OR spread OR control). We did not impose a publication date cutoff. See our protocol (Supplementary Note 1) and the guidance document for the production and collation of meta-analyses for the GLiTRS (GLobal Insect Threat-Response Synthesis) project[89] for further details on the search process, including how the search string was refined.

### Screening
Of the studies identified from our final search, those found in both databases were de-duplicated using remove_duplicates() in the litsearchr R package[90]. We then performed two rounds of screening on the resulting list of studies (Supplementary Data 1 provides full screening and exclusion details). In the first round, we screened the titles and abstracts only using the metagear R package[91] and discarded all studies that were irrelevant to our research question. For example, we discarded studies that discussed invasive alien species only as secondary factors and primarily focused on other anthropogenic threats such as urbanisation or land-use change (e.g., conversion to plantation), as attributing observed change to the presence of an invasive alien species is more complicated. We also excluded studies in which insects were considered only as the invasive alien species, rather than as taxa responding to the presence of invasive alien species. For the second screening round, we downloaded the full text of the

remaining studies and conducted a full-text screen based on our inclusion criteria (Supplementary Table 3). Briefly, for a study to be included, it needed to report the abundance, species richness, or biomass of native Hymenoptera, Coleoptera, Orthoptera, and/or Hemiptera in treatment (invasive alien species present) and control (invasive alien species absent) field sites. The data also needed to be reported to at least taxonomic order level and include summary statistics such as the mean, sample size, and a measure of variation, or provide sufficient primary data to calculate these values.

## Data extraction

The following data extraction processes were attempted in sequence; where one failed, we applied the next. First, wherever possible, the mean, sample size, and measure of variance for the treatment and control sites were extracted from tables in the main text or supplementary materials. Second, we used the shinyDigitise R package[92] to digitise data provided in graphical forms, such as a bar graph or scatter plot. Third, we used the raw data (if provided) to calculate the mean, sample size, and measure of variance. Lastly, we emailed the authors requesting access to their data.

During data extraction, we came across several scenarios where additional manipulation was required. First, where the authors reported a biodiversity measurement at the plot level (calculated by averaging multiple samples within each plot), we calculated a single biodiversity value for the invaded treatment and non-invaded control sites by calculating a weighted average of the plot-level means and the corresponding standard error, following the method described by Tatebe[93]. The weighted average of plot-level means $\bar{S}$ is calculated as

$$\bar{S} = \left(\frac{n_a}{n}\right)\bar{a} + \left(\frac{n_b}{n}\right)\bar{b} \tag{1}$$

where $n_a$ and $n_b$ are the sample sizes for plots $a$ and $b$, respectively; $n = n_a + n_b$ is the total sample size; and $\bar{a}$ and $\bar{b}$ are the plot-level means. The corresponding standard error $\varepsilon_S$ is calculated as

$$\varepsilon_S = \sqrt{\frac{N_a}{N}\varepsilon_a^2 + \frac{N_b}{N}\varepsilon_b^2 + \frac{n_a n_b (\bar{a} - \bar{b})^2}{nN}} \tag{2}$$

where $\varepsilon_a$ and $\varepsilon_b$ are the standard errors associated with plots $a$ and $b$, respectively; $N = (n^2 - n)$, $N_a = (n_a^2 - n_a)$, $N_b = (n_b^2 - n_b)$, and $n$, $n_a$, $n_b$, $\bar{a}$, and $\bar{b}$ are defined as above. Second, treatment and control sites were always defined as those where invasive aliens were present and absent, respectively, regardless of the description of the authors (e.g., a treatment where invasive alien species were removed).

To avoid duplication and pseudo-replication, we applied the following rules. First, we extracted data to the most refined taxonomic level available. For example, if a study reported results on Hymenoptera overall, and individual species such as *Bombus lapidaries* and *Andrena minutula*, we would extract the data for the individual species, and not include an additional data point for the order overall. Second, if a study reports results for multiple years, we only took the most recent data. Third, if a study reports results for multiple levels of invasion e.g., marginally invaded, moderately invaded, and extremely invaded, we only extracted the most extreme comparison (invasive alien species absent versus extremely invaded) to best reflect the definition of invasive alien species absent versus invasive alien species present.

Along with the mean, sample size, and variance measures, we extracted additional variables to serve as moderator variables for our second research question. To this end, we extracted year of publication, taxonomic description of the focal insect taxa, invasive alien species name, geographical realm (tropical if between 23 degrees north and 23 degrees south, otherwise non-tropical), whether the sites

were on an island smaller than 25,000 km², and whether the focal taxa could fly or not (ants were defined as non-flying). We were unable to extract data describing the intensity of the invasion since this information is typically not reported in a comparable or standardised way.

Spot checks were conducted at the study screening and data extraction stages by a second author. The second author screened 50 studies according to the same inclusion criteria and extracted data from five studies using the same data extraction spreadsheet. For the screening spot check, the calculated kappa statistic of 0.85 suggests very good agreement between the two authors. No concerning differences (e.g., strongly different values or different groups of values) were identified between the authors' sets of extracted data.

## Effect size calculation

Our dataset contains pairwise comparisons of the abundance, species richness, and biomass of Hymenoptera, Coleoptera, Hemiptera, and Orthoptera in sites with and without invasive alien species present. Any variance measures reported as standard error were converted to standard deviation before calculating the effect size for each pairwise comparison using the escalc() function from the metafor R package[94]. We chose the log response ratio (LRR) as our effect size due to its popularity in ecological meta-analyses for quantifying proportionate change and its robustness to non-independence[95,96]. A negative LRR indicates lower abundance, species richness, or biomass of the focal insect taxa when the invasive alien is present relative to a matching site in which the invasive alien is absent. An LRR close to zero indicates little effect relative to the control. As a high proportion of our extracted mean biodiversity measures were close to zero (Supplementary Fig. 5), we applied a bias correction to our effect sizes and associated variances using the delta method[97]. Accordingly, the adjusted effect sizes were calculated as

$$\text{Adjusted LRR} = \text{LRR} + \frac{1}{2}\left[\frac{(SD_T)^2}{N_T \bar{X}_T^2} - \frac{(SD_C)^2}{N_C \bar{X}_C^2}\right] \tag{3}$$

where $SD_T$ and $SD_C$ are the standard deviations, $N_T$ and $N_C$ are the sample sizes, and $\bar{X}_T$ and $\bar{X}_C$ are the mean biodiversity measures of the treatment and control groups, respectively. The adjusted variances were calculated as

$$\text{Adjusted } var = var + \frac{1}{2}\left[\frac{(SD_T)^4}{N_T^2 \bar{X}_T^4} + \frac{(SD_C)^4}{N_C^2 \bar{X}_C^4}\right] \tag{4}$$

To make the effect sizes more interpretable, we converted the adjusted LRRs to percentage changes[98]

$$\text{Percentage change} = 100 \times (e^{LRR} - 1) \tag{5}$$

where $e^{LRR}$ is the exponent of the log response ratio.

One effect size had substantially greater variance than all others (more than 25 times greater adjusted variance than the effect size with the second greatest adjusted variance), due to a relatively large standard deviation on a mean that was less than 0.1 (i.e., a poorly sampled insect species). This high variance effect size was removed before running any meta-analytic models.

## Meta-analytic models

We used the rma.mv() function from metafor[94] to run multi-level mixed-effects meta-analytic models for the estimation of a pooled effect size and 95% confidence intervals. The model specification was as follows

$$\text{metafor::rma.mv}(yi, vi, random = \sim 1|\text{Paper\_ID/Observation\_ID})$$

$$\tag{6}$$

where yi represents the effect size (LRR) for each individual observation and vi is the corresponding variance for each effect size. As effect sizes within a paper have a unique methodological context to which they relate, nested paper-level and observation-level random effects were used to account for non-independence within papers. Models were run separately for abundance and species richness, while biomass was not analysed due to too few effect sizes (1 study, 3 effect sizes). We considered invasive alien species to have a significant effect on insect biodiversity if the 95% confidence intervals of the overall model-derived effect did not overlap zero.

As is typical in an ecological meta-analysis, we expected high heterogeneity due to the differing contexts each effect size was collected under[54,99]. We quantified what proportion of this heterogeneity was due to within- and between-study differences using the var.comp() function from the dmetar R package[100], which provides multi-level $I^2$ (heterogeneity) estimations.

We ran further meta-analytic models to investigate variables likely to influence the direction or magnitude of the overall effect, using a series of meta-regression models with factors included as categorical predictors. These moderators included the year the study was published, the insect order of the focal taxon, whether the invasive alien species was a plant or animal, whether the data were collected in a tropical or non-tropical location, whether the data were collected from an island smaller than 25,000 km², and whether the focal taxon was known to fly. The metafor R package[94] provides the output of the QM test of moderators (an omnibus test) to indicate whether the included moderator explains a significant proportion of the heterogeneity, thus indicating there are differences between the groups. For the abundance data, we additionally tested a multi-moderator meta-regression including all moderators simultaneously. However, this approach reduced the sample size by 30%, and multicollinearity among moderators led to imprecise estimates. We therefore chose to model moderators separately—an approach commonly used in ecological meta-analyses[34,40], and one that reflects our aim to test distinct hypotheses for each moderator and assess whether they influence the direction or magnitude of the effect.

## Model sensitivity and publication bias checks

We took several steps to ensure confidence that our conclusions are supported by the evidence we present. Specifically, we followed the Koricheva and Gurevitch checklist[101] (Supplementary Note 2) for meta-analyses, meaning we used formal meta-analysis methodologies, clear documentation of the bibliographic search process, explicit inclusion and exclusion criteria, and thorough assessment of heterogeneity and potential bias.

To assess publication bias (i.e., whether studies with a particular effect have been selectively over- or under-published), we generated funnel plots to check for asymmetry. Additionally, we ran the rank correlation test with the ranktest() function from metafor[94] to formally assess funnel plot asymmetry. As an additional publication bias check, we implemented an adapted version of Egger's regression[102], which quantifies the relationship between effect sizes and their uncertainty, and is better suited than traditional Egger's regression and fail-safe numbers for datasets with non-independent effect sizes.

Finally, we conducted a series of sensitivity tests to assess whether using a different effect size metric or a certain subset of data changed the results. First, we re-ran the models with Hedge's g as the effect size instead of the LRR. Second, we ran our original models with only data points that passed Geary's test, defined as[97]

$$\frac{\bar{X}}{SD}\left(\frac{4N^{\frac{3}{2}}}{1+4N}\right) \geq 3 \tag{7}$$

where $\bar{X}$ is the mean, $SD$ is the standard deviation, and $N$ is the sample size of the biodiversity measure. For inclusion, both the treatment and control group must meet this rule based on their respective means, standard deviations, and sample sizes. Third, we ran the models on data points that are not disproportionately influential (as assessed by Cook's distance: data points were excluded if their Cook's distance exceeded 4/N). Fourth, although our focal taxa are typically terrestrial, some species within the focal orders are aquatic (such as Gerridae in Hemiptera). As our research question is primarily focused on the effect of invasive alien species on terrestrial insects, we also re-ran the models excluding any data that were collected via aquatic sampling methods (e.g., kick sampling). Lastly, we re-ran the abundance model incorporating a phylogenetic correlation matrix as a random effect to account for shared evolutionary history among taxa. After restricting the data to include only species-level data, we used the rotl R package[103] to import the phylogenetic data from the Open Tree of Life. We then used the ape R package[104] to apply Grafen's method (with the height argument set to its default value of 1) to estimate branch lengths, and to convert the tree to a correlation matrix for inclusion in the model.

Analyses were completed in R statistical software version 4.4.1[105]. We used multiple R packages for data preparation, analysis, and visualisation, including litsearchr 1.0.0[90], metagear 0.7[91], writexl 1.5.1[106], shinyDigitise 0.1.0[92], metafor 4.6-0[94], dmetar 0.1.0[100], tidyverse 2.0.0[107], rnaturalearth 1.0.1[108], waffle 1.0.2[109], ggimage 0.3.3[110], orchaRd 2.0[111], rotl 3.1.0[103], ape 5.8-1[104], stringr 1.5.1[112], Polychrome 1.5.4[113], and cowplot 1.1.3[114].

## Reporting summary

Further information on research design is available in the Nature Portfolio Reporting Summary linked to this article.

## Data availability

The data generated and analysed in this study have been deposited in a Zenodo repository (https://doi.org/10.5281/zenodo.14290020). Data from one contributing study are subject to data-sharing restrictions imposed by the data provider and therefore cannot be made publicly available[115]. Access to these data can be obtained by contacting the original data owner. Analyses using the shared dataset reproduce the reported results with only minor quantitative differences due to this omission. Supplementary Note 3 provides a complete list of references for all studies from which data were extracted for inclusion in the meta-analysis.

## Code availability

All code supporting this manuscript has been made publicly available on the Zenodo repository at https://doi.org/10.5281/zenodo.14290020.

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

## Acknowledgements

This work was funded by the Natural Environment Research Council as part of the GLiTRS (GLobal Insect Threat-Response Synthesis) project (grant numbers NE/V007548/1 [G.L.V.S., R.C., H.E.R., N.J.B.I., J.R.], NE/V006800/1 [J.M.], and NE/V006533/1 [CLO]). J.M. is also funded by the Leverhulme Trust and the Isaac Newton Trust on an Early Career Fellowship. H.E.R. is also funded by NC-UK (NE/Y006208/1).

## Author contributions

Conceptualisation: G.L.V.S., R.C., J.M., H.E.R., N.J.B.I., C.L.O., J.R. Data curation: G.L.V.S. Formal analysis: G.L.V.S. Methodology: G.L.V.S., R.C., J.M. Validation: G.L.V.S., R.C., J.M. Writing—original draft: G.L.V.S. Writing—review and editing: G.L.V.S., R.C., J.M., H.E.R., N.J.B.I., C.L.O., J.R. Project administration: G.L.V.S., J.M. Funding acquisition: N.J.B.I.

## Competing interests

The authors declare no competing interests.
