## [Transparent Peer Review file · Nature Communications]

Meta-analysis reveals negative but highly variable impacts of invasive alien species across terrestrial insect orders.

Corresponding Author: Ms Grace Skinner

Version 0:

Reviewer comments:

Reviewer #1

(Remarks to the Author)

This study is a meta-analysis focusing on the effects of invasive alien species on insect abundance and species richness. The authors found that both insect abundance and richness are lower in sites with alien species. The strongest effects are observed on the abundance of Hemiptera and the richness of Hymenoptera and alien animals had a stronger effect than alien plants. This is a very important study that enriches our understanding the impact of invasive alien species on insect communities. As the authors point out, insects are seldom considered the victims of invasions. I would also like to congratulate the authors for using a pre-registration for their study.

General comments

- 1) I would strongly recommend the authors to incorporate a phylogeny matrix in the meta-analytic models (this matrix can be constructed using the Open Tree of Life and the respective R packages ape and rotl and the matrix can be fitted in the model with the use of the argument "R" of the function rma.mv). More information about phylogenetic meta-analyses can be found here: <https://besjournals.onlinelibrary.wiley.com/doi/full/10.1111/2041-210X.13760>.
- 2) Somehow it is not clear throughout the manuscript that the category 'Animals' includes insects. Could you please clarify this in the Results (in the section 'Data description')?
- 3) Some important metrics are missing from the Results section, i.e. tau-squared (to quantify the variance among true effect sizes) and the Q test for heterogeneity (for moderators). Could you please add them?
- 4) It might be interesting to examine the effect of ants and dicotyledon plants with a subgroup analysis.

Specific comments

- 1) L83: Please replace the full stop with a (semi)colon after the phrase "research questions".
- 2) L91-93: Here there is a mix-up between two types of biases; publication bias and time-lag bias. The authors have examined both, but the phrasing here is not optimal.
- 3) L103: Please replace the full stop with ", while".
- 4) Figure 1C: What is shown in the axes? If the aim is to show just the relative frequencies of the different taxa, I would recommend a more complex figure that shows the distribution of invasive taxa per continent/country or per publication year.
- 5) L120: Rephrase to "The abundance of Hymenoptera, Coleoptera, Hemiptera, and Orthoptera was..."
- 6) L135, L146: Hedge's g is written with a capital H.
- 7) L187: Very nice idea to link the results to other threats.
- 8) L206: Could you please elaborate on the statement about competition?
- 9) L240-241: How do we know that the West Indian marsh created a favourable habitat for Coleoptera?
- 10) L228: context-dependent
- 11) L296: I think the lack of studies that report presence/absence is expected, because the search query excluded the keyword distribution. The reason for this is because presences and absences are the most commonly used data in species distribution models.
- 12) L324: Please rephrase to "PRISMA guidelines".
- 13) L357-358: I would move this Table to the Supplementary Material and just briefly mention the most important criteria in the text.
- 14) L406-436: The structure of this section is a bit confusing. Please re-structure so that the text presents the calculations that were done in a chronological order (e.g. the conversion to Percentage change probably took place after the calculation of the Adjusted LRR and its variance).

15) L456-459: This sentence is too long.

(Remarks on code availability)

The code is well structured. I was able to reproduce most of it.

Reviewer #2

(Remarks to the Author)

This meta-analysis is the first to quantitatively synthesize impacts of all types of invasive species on terrestrial insects. While previous studies have examined impacts of specific groups of invaders such as other terrestrial invertebrates, this more comprehensive assessment allows a better understanding of the vulnerability of insects to invasive alien species. The methods used in this manuscript are very clearly written and seem appropriate, and the results are also clearly described. I have fairly minor comments:

In the methods section, fail-safe numbers are used to assess publication bias, but I believe these are unable to handle non-independence among effect sizes and also don't control for heterogeneity (see <https://besjournals.onlinelibrary.wiley.com/doi/10.1111/2041-210X.13724>). The funnel plots that are also used are more appropriate, but it would be best to at least colour the points in the funnel plot so that effect sizes coming from the same study are the same colour (as suggested in the above paper). Also see <https://environmentalevidencejournal.biomedcentral.com/articles/10.1186/s13750-023-00301-6#Sec12> for another possible method (an extension of Egger's regression). Also in the methods, please add the citation for the R program itself.

Why were separate sub-group analyses chosen to look at moderators individually, versus a meta-regression approach, which apparently can be more powerful statistically (<https://environmentalevidencejournal.biomedcentral.com/articles/10.1186/s13750-023-00301-6#Sec8>; <https://link.springer.com/article/10.1007/s10682-012-9555-5>)? I know that sub-group analyses have been more common in the literature, but it seems like this study is well-suited for meta-regression.

It is not entirely clear to me if the 12 insect orders evaluated by Bladon et al. were other aquatic orders, or terrestrial orders that were not identified as being significantly threatened by invasive species. This should be clarified and potentially those orders should be listed there, since it isn't possible for readers to refer to the manuscript in preparation.

Reference 91 (Tatebe) seems incomplete, and it would be better to give more explanation on this method in the text as well.

In Figure 1b, there should be a scale indicating what the circle size means (presumably it is proportional to the number of studies, but what size equates to what number of studies?)

(Remarks on code availability)

I was able to install and run the code, and reproduce the results for the models. The dmetar package did not seem to be available for the latest version of R, and therefore I did not manage to run the var.comp function to calculate the I2 but presumably it would work with an earlier version.

Version 1:

Reviewer comments:

Reviewer #2

(Remarks to the Author)

The responses to both of the reviewers are thorough and I do not see any further issues in the revised manuscript. I also ran some of the new code and did not encounter any problems.

(Remarks on code availability)

Reviewer #3

(Remarks to the Author)

This meta-analysis is a great contribution to our understanding of insect declines! It is well performed and documented, and the authors seem to have properly addressed most previous referees concerns. I believe the statistical analysis is sound, and the authors have properly considered potential bias and non-independence. I have just a couple of comments.

First, I am concerned about their restriction of the taxa included. I understand that they were focusing on taxa more likely to be affected by invasive species based on an unpublished manuscript. But I cannot access the manuscript and I cannot know how these taxa were assessed. Not being able to evaluate the manuscript used to exclude other taxa is relatively minor. A main concern for me is that they are excluding two very abundant and rich orders: Lepidoptera and Diptera. These two orders together can account for about 300,000 species, whereas Orthoptera and Hemiptera together account for about

100,000 species. Thus, because they are not including almost a third of the insects, I believe the authors cannot say this is a test of the effect of invasive species on terrestrial insects. And they also did not find a negative effect on Coleoptera, the most diverse insect order. Lastly, by focusing on orders more likely to be affected by invasive species, they might be more prone to finding negative effects. This restriction could be by itself a bias in their study. I suggest toning down some of the main conclusions (e.g., line 332 and title).

Second, I believe this meta-analysis is novel in focusing solely on insects and including all types of invasive species, but there are other meta-analyses that have evaluated the effect of invasive species on insects. I found at least four meta-analyses that consider the impact of invasives on animals (not just insects, but at least a good portion of the taxa evaluated are insects or they have subdivisions by taxon), and one specific about Lepidoptera. These meta-analyses should be considered to be included in the paragraph starting in line 56 (and maybe might be useful for the rest of the manuscript as well):

Charlebois, J. A., & Sargent, R. D. (2017). No consistent pollinator-mediated impacts of alien plants on natives. *Ecology Letters*, 20(11), 1479-1490.

Fletcher, R. A., Brooks, R. K., Lakoba, V. T., Sharma, G., Heminger, A. R., Dickinson, C. C., & Barney, J. N. (2019). Invasive plants negatively impact native, but not exotic, animals. *Global Change Biology*, 25(11), 3694-3705.

Yoon, S. A., & Read, Q. (2016). Consequences of exotic host use: impacts on Lepidoptera and a test of the ecological trap hypothesis. *Oecologia*, 181(4), 985-996.

Montero-Castaño, A., & Vilà, M. (2012). Impact of landscape alteration and invasions on pollinators: a meta-analysis. *Journal of Ecology*, 100(4), 884-893.

Schirmel, J., Bundschuh, M., Entling, M. H., Kowarik, I., & Buchholz, S. (2016). Impacts of invasive plants on resident animals across ecosystems, taxa, and feeding types: a global assessment. *Global change biology*, 22(2), 594-603.

(Remarks on code availability)
Code seems okay.

Response to reviewers

Dear Reviewer #1, and Reviewer #2,

Thank you very much for your feedback and comments on our manuscript, 'Meta-analysis reveals negative but highly variable impacts of invasive alien species on terrestrial insects'. We agree with the revisions recommended and have made the corresponding changes, which we believe have strengthened our meta-analysis.

Overall, we have:

- Conducted additional models and tests to improve the robustness of our results by accounting for non-independence in the data. For example, we explored whether incorporating a phylogeny matrix as a random effect improved model fit and replaced the fail-safe number test with an adapted version of Egger's regression to assess publication bias, due to its superior ability to handle non-independent effect sizes.
- Provided further clarification on key concepts to improve readers' understanding and interpretation of our analyses. Specifically, we clarified that the 'Animals' invasive category includes insects, more clearly described the types of bias investigated and the models employed, explained our rationale for modeling moderators separately, and explicitly reported additional metrics for variance and heterogeneity.
- Improved the manuscript's clarity and readability by moving some methodological details to the supplementary materials, and updating figures to highlight key messages more effectively.

Specific changes are detailed in blue text below each corresponding point of feedback. We include line numbers corresponding to the revised manuscript with tracked changes on, along with screenshots to locate the relevant changes.

Thank you for these valuable recommendations which have helped to improve our manuscript.

On behalf of all authors,
Grace Skinner

Reviewer #1

Remarks to the Author

This study is a meta-analysis focusing on the effects of invasive alien species on insect abundance and species richness. The authors found that both insect abundance and richness are lower in sites with alien species. The strongest effects are observed on the abundance of Hemiptera and the richness of Hymenoptera and alien animals had a stronger effect than alien plants. This is a very important study that enriches our understanding the impact of invasive alien species on insect communities. As the authors point out, insects are seldom considered the victims of invasions. I would also like to congratulate the authors for using a pre-registration for their study.

Thank you for taking the time to comment on our manuscript and for your positive feedback. We address each of your comments below in turn.

General comments

1) I would strongly recommend the authors to incorporate a phylogeny matrix in the meta-analytic models (this matrix can be constructed using the Open Tree of Life and the respective R packages ape and rotl and the matrix can be fitted in the model with the use of the argument "R" of the function rma.mv). More information about phylogenetic meta-analyses can be found here:

<https://besjournals.onlinelibrary.wiley.com/doi/full/10.1111/2041-210X.13760>

We now also test rma.mv model accounting for phylogeny, using the rotl and ape packages to construct a tree. Given we were only able to construct a tree for a reduced set of effect sizes (only 40% of the data points are at species-level) this model was performed on a filtered subset. We show that for this reduced set of effect sizes, a model without the phylogenetic structure had the lower AIC value (AIC = 385.8 with phylogeny matrix vs AIC = 384.7 without). To clarify for the reader, we now introduce this sensitivity test in the methods (L577) and refer to it in the results (L149-150).

577 sampling methods (e.g., kick sampling). Lastly, we re-ran the abundance model incorporating a
578 phylogenetic correlation matrix as a random effect to account for shared evolutionary history
579 among taxa. After restricting the data to include only species-level data, we used the rotl R
580 package⁹⁸ to import the phylogenetic data from the Open Tree of Life. We then used the ape R

2)

32

581 package⁹⁹ to apply Grafen's method (with the height argument set to its default value of 1) to
582 estimate branch lengths, and to convert the tree to a correlation matrix for inclusion in the model.

corrected standardized mean of either the treatment or control did not pass Geary's test. The AIC
value was greater for a model including a phylogenetic correlation matrix as a random effect
(Supplementary Table 1)(Table S1).

Somehow it is not clear throughout the manuscript that the category 'Animals' includes insects. Could you please clarify this in the Results (in the section 'Data description')?

We have added a clarification in the 'Data description' section stating that invasive alien animals include insects (L111).

Regarding the invasive alien species investigated, 30 studies assessed an invasive alien animal
(including insects), while 22 assessed an invasive alien plant. Ants and dicotyledon plants were

3) Some important metrics are missing from the Results section, i.e. tau-squared (to quantify the variance among true effect sizes) and the Q test for heterogeneity (for moderators). Could you please add them?

Thank you for raising this, we agree that these statistics are important to include. For our multilevel models, heterogeneity was partitioned into variance in effect sizes both between studies and within

studies. We now report this variance between and within studies, along with the results of the QE tests for residual heterogeneity for the models with moderators (L129-136, L179-183, L191-195).

with invasive alien species (Fig. 2). Heterogeneity for abundance data, assessed with multi-level
I^2 , indicates high variation (91%), with ~~within~~between-study differences accounting for ~~2863~~% of
the variation, and ~~between~~within-study differences accounting for ~~6328~~%. The variance among
true effect sizes for abundance was partitioned into variance between studies ($\sigma^2 = 0.32$) and
within studies ($\sigma^2 = 0.70$). For species richness data, heterogeneity was 98% (37% between, 60%
within). The variance components for species richness were $\sigma^2 = 0.23$ between studies and $\sigma^2 =$
0.37 within studies. The equivalent figures for species richness data were 98% heterogeneity
(60% within, 37% between).

invasive aliens by different insect orders. The variance among true effect sizes for abundance was
partitioned into components between studies ($\sigma^2 = 0.33$) and within studies ($\sigma^2 = 0.67$). For
species richness, the corresponding values were $\sigma^2 = 0.19$ and $\sigma^2 = 0.34$. Residual heterogeneity
was significant for both abundance (QE = 2008.72, ~~df~~ = 273, $p < 0.0001$) and species richness (QE
= 2368.89, ~~df~~ = 37, $p < 0.0001$), indicating that unaccounted variation remains in both models.

invasive plant group was dominated by dicotyledon plants (108 effect sizes, 69%) (Fig. 1C). For
abundance, variance among true effect sizes was partitioned into components between studies (σ^2
= 0.26) and within studies ($\sigma^2 = 0.71$). The corresponding values for species richness were $\sigma^2 =$
0.22 and $\sigma^2 = 0.38$. Significant residual heterogeneity remained in both models (abundance: QE =
2153.6, ~~df~~ = 275, $p < 0.0001$; species richness: QE = 3018.45, ~~df~~ = 38, $p < 0.0001$).

4) It might be interesting to examine the effect of ants and dicotyledon plants with a subgroup analysis.

Thank you for the suggestion to examine the effects of invasive ants and dicotyledon plants separately. Given that ants and dicots comprised the majority of the data in the broader categories of invasive animals and plants, respectively, we found that additional subgroup analyses yielded very similar results to our original analysis using the broader animal vs. plant categorisation: ants significantly reduced

native insect abundance, while dicots did not (see figure below). As such, we have decided not to include this additional result to avoid redundancy. However, we have updated the main text to note that the majority of effect sizes for invasive plants were dicots, building on the original statement that most invasive animals were ants (L190-191).

Abundance effect sizes

significantly differ (QM = 0.9556, $p = 0.3283$). Most of the effect sizes for invasive animals
 described the effect of invasive alien insects, particularly ants (123 effect sizes, 76%), while the
 invasive plant group was dominated by dicotyledon plants (108 effect sizes, 69%)-(Fig. 1C). For

Specific comments

1) L83: Please replace the full stop with a (semi)colon after the phrase “research questions”.

We have replaced the full stop with a colon (L85).

preparation]). We address two key research questions: 1. What is the impact of invasive alien

2) L91-93: Here there is a mix-up between two types of biases; publication bias and time-lag bias. The authors have examined both, but the phrasing here is not optimal.

Reviewer 1 is correct that we tested for two forms of bias here. We include a moderator for year of publication, grouped into four time periods, to test the extent to which publication date predicts the reported effect of invasive alien species. Second, we visually inspected funnel plots, performed a rank

correlation test, and implemented an adapted version of Egger's regression. We clarify for the reader our description of the year of publication moderator at the end of the introduction (L93-94), and then refer the reader to our tests for publication bias in the methods (L549-555).

additionally examine the year of study publication as a potential moderator to investigate the
extent to which publication date predicts the reported effect of invasive alien species whether
earlier studies reported stronger effects, reflecting a potential bias towards faster publication of
more significant results⁴⁶.

To assess publication bias (i.e., whether studies with a particular effect have been selectively over-
or under-published), we generated funnel plots to check for asymmetry. Additionally, we ran the
rank correlation test with the ~~ranktest()~~ function from metafor⁸⁹ to formally ~~check assess for~~ funnel
plot asymmetry. As an additional publication bias check, we implemented an adapted version of
Egger's regression⁹⁷, which quantifies the relationship between effect sizes and their uncertainty,
and is better suited than traditional Egger's regression and fail-safe numbers for datasets with
non-independent effect sizes. ~~We also calculated Rosenthal's fail safe number using the fsn()~~

3) L103: Please replace the full stop with “, while”.

We have replaced this full stop with “, while” (L107).

originate from North America (n = 81; 25%) and Europe (n = 105; 33%), reflecting wider spatial
biases in insect data^{47,48}, while, 16% of effect sizes originated from tropical biomes, and 7% from
islands.

4) Figure 1C: What is shown in the axes? If the aim is to show just the relative frequencies of the different taxa, I would recommend a more complex figure that shows the distribution of invasive taxa per continent/country or per publication year.

Figure 1C is a waffle plot representing our 318 effect sizes, with each rectangle corresponding to one effect size. We now clarify this for the reader both on the figure and in the legend. For years of publication and the geographic distribution of effect sizes, we refer the reader to panels 'a' and 'b'.

C

Figure. 1. Temporal, spatial, and taxonomic coverage of collated data. Frequency of 318 insect biodiversity effect sizes collected according to time, geography, taxonomy of invasive alien species, and taxonomy of focal insect taxa. (Aa) The temporal distribution

of effect sizes, showing the cumulative number of effect sizes from 1995 to 2022; (bB) the global distribution of effect sizes. Colour indicates the taxonomic order of the focal taxa; Coleoptera (green), Hemiptera (pink), Hymenoptera (purple), Orthoptera (orange); size indicates number of effect sizes. Landmass polygons from the `naturalearth` R package, displayed using a Mollweide projection; (Cc) waffle plot showing effect sizes split by the taxonomic distribution of the invasive alien species. Each rectangle represents one effect size. The taxonomic categories are: insects (orange), plants (green), vertebrates (purple), non-insect invertebrates (blue); (Dd) effect sizes split by the taxonomic distribution of the focal insect taxa.

5) L120: Rephrase to “The abundance of Hymenoptera, Coleoptera, Hemiptera, and Orthoptera was...”

We have rephrased this sentence according to your suggestion (L125).

~~The abundance of For insects in the orders~~ Hymenoptera, Coleoptera, Hemiptera, and Orthoptera,
~~abundance~~ was 31% lower on average (95% confidence interval: 45% to 14% lower; LRR: -0.37

6) L135, L146: Hedge’s g is written with a capital H.

We have capitalised Hedge’s g (L146, L160).

qualitatively change under multiple sensitivity analyses, including when using ~~h~~Hedge’s g as the
excluding data collected with aquatic sampling techniques, but not when using ~~h~~Hedge’s g as the

7) L187: Very nice idea to link the results to other threats.

Thank you for the positive feedback!

8) L206: Could you please elaborate on the statement about competition?

We have revised the sentence to clarify that the greater impact of invasive alien animals on native insects may be due to more direct competition for similar resources, compared to the more indirect effects of invasive alien plants, which may take longer to become apparent (L234-237).

and species richness than invasive alien plants. This greater impact may be due to ~~their more~~
direct competition ~~between with~~ native insects ~~and invasive alien animals for similar resources,~~
~~compared to the more indirect effects of invasive alien plants, for similar niches,~~ leading to more
immediate ~~and direct~~ effects⁵⁶. Our findings are consistent with Terceel et al.³⁴, who focused on the

9) L240-241: How do we know that the West Indian marsh created a favourable habitat for Coleoptera?

The statement about West Indian marsh grass creating a favorable habitat for Coleoptera is based on the findings of Houston and Duivenvoorden (2002), who observed higher numbers of Coleoptera in invaded beds compared to native plant beds. While this suggests that the habitat may be more favorable for Coleoptera, we acknowledge that this is a correlation, and we have adjusted the

manuscript accordingly to reflect this. We have added an additional citation for the Houston and Duivenvoorden (2002) study to provide further clarity. (L272-273).

Hemiptera. Notably, while West Indian marsh grass negatively affected Hemiptera, it appears to
created a more favourable habitat for Coleoptera⁶⁶. Interestingly, Tercei et al.³⁴ found that

10) L228: context-dependent

We have hyphenated 'context-dependent' (L259).

impacts ~~of invasive alien species~~⁵⁹. Indeed, the outcomes of biological invasions are highly
context-dependent⁶⁰. In novel ecosystems, where native vegetation has been lost, alien plants

11) L296: I think the lack of studies that report presence/absence is expected, because the search query excluded the keyword distribution. The reason for this is because presences and absences are the most commonly used data in species distribution models.

We have included a statement to explain that the lack of studies reporting incidence is likely due to the exclusion of the keywords 'incidence' and 'distribution' in our search (L331-333).

rate; could provide more insights into the effect of invasive aliens. The lack of studies reporting
incidence (presence/absence) is likely due to the exclusion of the keywords 'incidence' and
'distribution' in our search. ~~though they are unlikely to alter the overall conclusion that invasive~~

12) L324: Please rephrase to "PRISMA guidelines".

We have rephrased to "PRISMA guidelines" (L360).

Following ~~the~~ PRISMA guidelines methodology^{82,83}, we collated studies assessing the impact of

13) L357-358: I would move this Table to the Supplementary Material and just briefly mention the most important criteria in the text.

We agree that this level of detail is not essential to the main text, so we have moved the table with the full inclusion criteria to the Supplementary Material (Table S2) and provided a concise summary of the key criteria in the main text (L397-402).

remaining studies and conducted a full-text screen based on our inclusion criteria (Supplementary
 Table 24). Briefly, for a study to be included, it needed to report the abundance, species richness,
 or biomass of native Hymenoptera, Coleoptera, Orthoptera, and/or Hemiptera in treatment
 (invasive alien species present) and control (invasive alien species absent) field sites. The data
 also needed to be reported to at least taxonomic order level and include summary statistics such as
 the mean, sample size, and a measure of variation, or provide sufficient primary data to calculate
 these values.

 ~~**Table 1. Study inclusion criteria. Inclusion criteria for studies assessing the impact of invasive**~~
 ~~**alien species on native insect orders.**~~

Inclusion criterion	Description
Focal insect orders	The study reports the impact of invasive alien species on native Hymenoptera, Coleoptera, Orthoptera, and/or Hemiptera.
Anthropogenic	The study focuses on invasive alien species (9.1 Invasive non-native/alien

14) L406-436: The structure of this section is a bit confusing. Please re-structure so that the text presents the calculations that were done in a chronological order (e.g. the conversion to Percentage change probably took place after the calculation of the Adjusted LRR and its variance).

We have rearranged this section such that we now first introduce the equation and text for the calculation of the adjusted log response ratio, and then the conversion of that adjusted log response ratio to a percentage change (L495-499).

*Effect size calculation*

Our datasetbase contains pairwise comparisons of the abundance, species richness, and biomass
of Hymenoptera, Coleoptera, Hemiptera, and Orthoptera in sites with and without invasive alien
species present. Any variance measures reported as standard error were converted to standard
deviation before calculating the effect size for each pairwise comparison using the escalc()
function from the metafor R package⁸⁹. We chose the log response ratio (LRR) as our effect size
due to its popularity in ecological meta-analyses for quantifying proportionate change and its
robustness to non-independence^{90,91}. The LRR quantifies proportionate change between
treatments and can be converted to an easily interpretable percentage change⁹³

(4)
where \$e^{LRR}\$ is the exponent of the log response ratio. A negative LRR indicates lower abundance,

variances were calculated as

$$492 \text{ Adjusted var} = \text{var} + \frac{1}{2} \left[\frac{(SD_T)^4}{N_T^2 \bar{X}_T^4} + \frac{(SD_C)^4}{N_C^2 \bar{X}_C^4} \right]$$

(43)

To make the effect sizes more interpretable, we converted the adjusted LRRs to percentage
change⁹³

$$497 \text{ Percentage change} = 100 * (e^{LRR} - 1)$$

(5)

where \$e^{LRR}\$ is the exponent of the log response ratio.

One effect size had substantially greater variance than all others (more than 25 times greater

15) L456-459: This sentence is too long.

We have revised the sentence by breaking it into two sentences (L525-532).

We ran further meta-analytic models to investigate variables ~~we thought~~ likely to influence the
~~direction or magnitude of the overall effect, size of the effect by including them as moderator~~
~~variables. using a series of meta-regression models~~~~We investigated whether~~ with factors included
as categorical predictors. These moderators included the year the study was published, the insect
order ~~of that~~ the focal tax~~ona belonged to~~, whether the invasive alien species was a plant or
animal, whether the data were collected ~~in from~~ a tropical or non-tropical location, whether the
data were collected from an island smaller than 25,000 km², and whether the focal tax~~ona~~ was

30

known to fly. ~~influenced the direction or magnitude of the overall effect size.~~ The metafor R

Remarks on code availability

The code is well structured. I was able to reproduce most of it.

This is good to hear!

Reviewer #2

Remarks to the Author

This meta-analysis is the first to quantitatively synthesize impacts of all types of invasive species on terrestrial insects. While previous studies have examined impacts of specific groups of invaders such as other terrestrial invertebrates, this more comprehensive assessment allows a better understanding of the vulnerability of insects to invasive alien species. The methods used in this manuscript are very clearly written and seem appropriate, and the results are also clearly described. I have fairly minor comments:

Thank you for taking the time to comment on our manuscript and for your positive feedback. We address each of your comments below in turn.

In the methods section, fail-safe numbers are used to assess publication bias, but I believe these are unable to handle non-independence among effect sizes and also don't control for heterogeneity (see <https://besjournals.onlinelibrary.wiley.com/doi/10.1111/2041-210X.13724>). The funnel plots that are also used are more appropriate, but it would be best to at least colour the points in the funnel plot so that effect sizes coming from the same study are the same colour (as suggested in the above paper). Also see <https://environmentalevidencejournal.biomedcentral.com/articles/10.1186/s13750-023-00301-6#Sec12> for another possible method (an extension of Egger's regression). Also in the methods, please add the citation for the R program itself.

Thank you for this helpful suggestion. We have now removed the fail-safe number analysis, and instead implemented an adapted Egger's regression to handle the non-independent effect sizes. This change is now reflected in both the Methods (L552-555) and Results (L137-144, L153-156) sections of the manuscript.

plot asymmetry. As an additional publication bias check, we implemented an adapted version of
Egger's regression⁹⁷, which quantifies the relationship between effect sizes and their uncertainty,
and is better suited than traditional Egger's regression and fail-safe numbers for datasets with
non-independent effect sizes. We also calculated Rosenthal's fail safe number using the fsn()
function from metafor⁹¹, which estimates how many unpublished, non-significant results would

31

be required to change the result. Generally, the result is considered robust if the fail safe number
is greater than $5N+10$, where N is the total number of studies. Therefore, the fail safe numbers
for the abundance and species richness models should be greater than 255 and 130 studies,
respectively.

For the abundance models, the funnel plots were visually symmetrical (Supplementary Fig. S2)
around the overall effect size, showing no apparent publication bias. The rank correlation test
(non-significant asymmetry; Kendall's tau = -0.0142, p = 0.7245) and adapted Egger's regression
(no relationship between effect size and its error; estimate = 0.2463, p = 0.6112) formally
supported this, indicating no concerns of publication bias (non-significant asymmetry; Kendall's
tau = -0.0142, p = 0.7245). Still, the data points did not form the classic funnel shape, likely due
to high heterogeneity across ecological studies, where larger studies do not necessarily show
greater precision^{46,49,50}. Nevertheless, Rosenthal's fail safe number (23945) greatly exceeded the
minimum threshold of 255, indicating that the analysis was robust. Our results did not

The rank correlation test for the species richness model indicated potential funnel plot asymmetry
there could be publication bias (Kendall's tau = -0.2872, p = 0.0088), although this was not
supported by the adapted Egger's regression (estimate = 0.7596, p = 0.3140), which found no
relationship between effect size and its error. ~~Rosenthal's fail safe number (2829) was greater~~
~~than the minimum required threshold of 130, indicating non-significant asymmetry and thus a~~
~~lack of strong publication bias.~~ For the sensitivity analyses, the results were less consistent: the

We have also updated the funnel plots in the Supplementary Materials so that points are coloured by study, and included the number of studies in the figure headings.

Supplementary Figure 2. Funnel plots used to assess publication bias. Effect sizes (delta corrected Log Response Ratio [LRR]) against the standard error describing the impact of invasive alien species presence on terrestrial insect (Coleoptera, Hemiptera, Hymenoptera, Orthoptera) (a) abundance and (b) species richness. Points are coloured by study.

We also apologise for the oversight in not citing the R program itself. We have added this now (L584).

Analyses were completed in R statistical software version 4.4.1¹⁰⁰. We used multiple R packages

Why were separate sub-group analyses chosen to look at moderators individually, versus a meta-regression approach, which apparently can be more powerful statistically (<https://environmentalevidencejournal.biomedcentral.com/articles/10.1186/s13750-023-00301-6#Sec8>; <https://link.springer.com/article/10.1007/s10682-012-9555-5>)? I know that sub-group analyses have been more common in the literature, but it seems like this study is well-suited for meta-regression.

Thank you for this constructive comment. To clarify, our analyses were conducted using meta-regression models. Moderators (e.g., invasive alien type, insect taxonomic order) were included as categorical predictors in meta-regression models, rather than running independent models on subgroups of data and comparing the results of the subgroups using statistical tests (as in traditional subgroup analysis). This follows the approach described in the paper the reviewer cited, where the authors note that: “Traditionally, researchers conduct separate meta-analyses per different groups (known as ‘sub-group analysis’), but we prefer a meta-regression approach with a categorical variable, which is statistically more powerful [40].” We have clarified the description of our approach in the manuscript by explicitly stating that meta-regression models were used, and that we ran a series of meta-regression models with our moderator variables (L525-528).

We ran further meta-analytic models to investigate variables ~~we thought~~ likely to influence the
direction or magnitude of the overall effect, size of the effect by including them as moderator
variables. using a series of meta-regression models~~We investigated whether~~ with factors included
as categorical predictors. These moderators included the year the study was published, the insect

We recognise the reviewer may be suggesting a multi-moderator meta-regression, where all moderators are included simultaneously. We explored this for our dataset.

For the abundance data, including all moderators in one model reduced the sample size by 30% because flight capability data was not available for 84 of the 277 effect sizes. Additionally, there were issues with non-independent moderators and unbalanced moderator levels. For example, only 20 data points came from islands compared to 257 from mainlands, and 18 of the 20 data points originating from islands were also from tropical biomes. We found substantial multicollinearity, with variance inflation factors (VIFs) exceeding 3 for half of the parameters and reaching up to 8. This likely contributed to the imprecise estimates, where standard errors were often larger than the estimates themselves.

For species richness, while the multi-moderator model converged, the sample size ($n = 40$) was insufficient to robustly estimate the large number of parameters, falling well below recommended thresholds of including at least 10 studies per moderator modelled (Cochrane handbook section 10.11.5.1 <https://www.cochrane.org/authors/handbooks-and-manuals/handbook/current/chapter-10#section-10-11-5>). Even in the single moderator models, the flight capability, island/mainland, tropical/non-tropical, and year moderators were not analysed due to very limited data for at least one level of each moderator.

Given these issues, we chose to present the single meta-regression models for both abundance and species richness in the main text, an approach widely used in ecological meta-analyses (Tercel et al. (2023); Wang et al. (2020)), especially when datasets are limited or contain missing moderator data. For transparency, we include the results of the multi-moderator abundance model in the Supplementary

Material (Supplementary Table 3). We also introduce the approach in the methods (L535-540) and comment on the findings in the results (L200-203).

are differences between the groups. For the abundance data, we additionally tested a multi-
 moderator meta-regression including all moderators simultaneously. However, this approach
 reduced the sample size by 30%, and multicollinearity among moderators led to imprecise
 estimates. We therefore chose to model moderators separately—an approach commonly used in
 ecological meta-analyses^{34,37}, and one that reflects our aim to test distinct hypotheses for each
 moderator and assess whether they influence the direction or magnitude of the effect.^{34,37}

moderator did not differ from one another. While a multi-moderator model produced some
 differing results, due to reduced sample size and variance inflation factors (VIFs) indicating
 multicollinearity among some moderators, we had less confidence in those estimates
 (Supplementary Table 3). -These moderators were not assessed in relation to species richness due

Supplementary Table 3. Results of the multi-moderator meta-regression model using the abundance dataset. Output of an abundance model including all moderators simultaneously (k = 193). Estimates represent the difference in effect size for each moderator level relative to the baseline (reference) level of each moderator, controlling for all other moderators in the model.

Moderator	Estimate (95% confidence intervals)	Standard error
Intercept	-0.0163 (-2.5838, 2.5512)	1.3100
Order: Hemiptera	-1.4891 (-2.2849, -0.6932)	0.4061
Order: Hymenoptera	-0.0046 (-0.6329, 0.6238)	0.3206
Order: Orthoptera	-1.3549 (-3.4258, 0.7160)	1.0566
Invasive type: plant	0.3109 (-0.4907, 1.1124)	0.4090
Year group: 2000-2010	-0.7087 (-2.1323, 0.7150)	0.7264
Year group: 2011-2020	-0.5906 (-1.9786, 0.7975)	0.7082
Year group: post-2020	-1.9965 (-3.8370, -0.1561)	0.9390
Biome: Tropical	1.1394 (-0.6618, 2.9405)	0.9190
Island or mainland: Mainland	0.8077 (-1.3393, 2.9547)	1.0954
Flight capability: Non-flying	-0.6811 (-1.2846, -0.0775)	0.3079

It is not entirely clear to me if the 12 insect orders evaluated by Bladon et al. were other aquatic orders, or terrestrial orders that were not identified as being significantly threatened by invasive species. This should be clarified and potentially those orders should be listed there, since it isn't possible for readers to refer to the manuscript in preparation.

Thank you for highlighting that this was not clear. We have now listed all 12 insect orders evaluated in Bladon et al., clarified which were identified as being significantly threatened by invasive alien species, and specified the four primarily terrestrial orders from this group that formed the focus of our meta-analysis (L365-372).

potential threat (Bladon et al. [manuscript in preparation]). ~~Note~~This assessment evaluated 12
insect orders (Lepidoptera, Hymenoptera, Coleoptera, Diptera, Phasmatodea, Orthoptera,

21

~~Hemiptera, Dermaptera, Odonata, Ephemoptera, Plecoptera, and Trichoptera)~~ were evaluated by
~~Bladon et al.~~, representing 96% of described insect species. Of these, invasive alien species were
ranked among the top 10 threats for Hymenoptera, Coleoptera, Orthoptera, Hemiptera, Odonata,
Ephemoptera, Plecoptera, and Trichoptera. For this meta-analysis, we focused on the four
primarily terrestrial orders from this group: Hymenoptera, Coleoptera, Orthoptera, and
Hemiptera.

Reference 91 (Tatebe) seems incomplete, and it would be better to give more explanation on this method in the text as well.

Thank you for pointing this out. Reference 91 is not a standard journal article, and the formatting was incorrect as a result. We have now updated both the in-text citation and the reference in the bibliography. We have expanded the explanation in the main text by explicitly including the relevant equations from the source (L421-436).

required. First, where the authors reported a biodiversity measurement at the plot level (calculated
 by averaging multiple samples within each plot), ~~we combined the averaged plot level data to~~
 ~~obtain a single biodiversity measurement for the invaded treatment and non-invaded control site,~~
 ~~following the methods of Tatebe~~ we calculated a single biodiversity value for the invaded
 ~~treatment and non-invaded control sites by calculating a weighted average of the plot-level means~~
 ~~and the corresponding standard error, following the method described by Tatebe (2005)⁸⁸. The~~
 ~~weighted average of plot-level means \bar{S} is calculated as~~

$$\bar{S} = \left(\frac{n_a}{n}\right)\bar{a} + \left(\frac{n_b}{n}\right)\bar{b}$$

(1)

~~where n_a and n_b are the sample sizes for plots a and b , respectively; $n = n_a + n_b$ is the total~~
 ~~sample size; and \bar{a} and \bar{b} are the plot-level means. The corresponding standard error ϵ_s is~~
 ~~calculated as~~

$$\epsilon_s = \sqrt{\frac{N_a}{N} \epsilon_a^2 + \frac{N_b}{N} \epsilon_b^2 + \frac{n_a n_b (\bar{a} - \bar{b})^2}{nN}}$$

26

(2)

~~where ϵ_a and ϵ_b are the standard errors associated with plots a and b , respectively; $N =$~~
 ~~$(n^2 - n) \cdot N_a = (n_a^2 - n_a) \cdot N_b = (n_b^2 - n_b)$, and $n \cdot n_a \cdot n_b \cdot \bar{a}$, and \bar{b} are defined as above.~~

In Figure 1b, there should be a scale indicating what the circle size means (presumably it is proportional to the number of studies, but what size equates to what number of studies?)

We have added an additional legend to Figure 1b to indicate how many studies each circle represents.

b

Figure 1. Temporal, spatial, and taxonomic coverage of collated data. Frequency of 318 insect biodiversity effect sizes collected according to time, geography, taxonomy of invasive alien species, and taxonomy of focal insect taxa. (Aa) The temporal distribution

of effect sizes, showing the cumulative number of effect sizes from 1995 to 2022; (bB) the global distribution of effect sizes. Colour indicates the taxonomic order of the focal taxa; Coleoptera (green), Hemiptera (pink), Hymenoptera (purple), Orthoptera (orange); size indicates number of effect sizes. Landmass polygons from the `naturalearth` R package,

Remarks on code availability

I was able to install and run the code, and reproduce the results for the models. The `dmetar` package did not seem to be available for the latest version of R, and therefore I did not manage to run the `var.comp` function to calculate the I2 but presumably it would work with an earlier version.

Thank you for testing the code and confirming the results. Apologies that you were unable to run the `var.comp` function from the `dmetar` package. We were able to install `dmetar` using the instructions at <https://dmetar.protectlab.org/> on R version 4.4.2. However, we've noted that R version 4.5.0 is currently being rolled out, which may not yet support the package.

Response to reviewers (second revision)

Dear Reviewers,

We thank the reviewers for their time and helpful feedback on our manuscript, 'Meta-analysis reveals negative but highly variable impacts of invasive alien species on terrestrial insects'. We understand that Reviewer #1 was not available to provide further comments in this round. We appreciate the positive feedback from Reviewer #2 and the constructive comments from Reviewer #3, and we have addressed all their comments in detail below.

Overall, we have:

- Refined our conclusions to better reflect the study's taxonomic scope. We have more explicitly stated throughout the manuscript – including in the abstract and first sentence of the Discussion – that our findings apply to the subset of terrestrial insect orders included in the study.
- Expanded the Discussion to note that broader inclusion of insect taxa (beyond those orders identified by experts as having invasive alien species among their top threats) might reveal more variable and less negative responses.
- Replaced reference to an unpublished manuscript (Bladon et al.) with citation of a now-published paper (Millard et al., 2025; <https://doi.org/10.1111/ddi.70025>) that presents the referenced material, explaining some of our rationale and approach for the expert ranking of threats.
- Placed greater emphasis on the conclusion of 'highly variable impacts' in key summary paragraphs; and cited existing literature where broader animal-focused meta-analyses (which included, but did not focus on, insects) reported similarly negative but variable effects.
- Added additional citations to relevant meta-analyses to provide further context when introducing our study.

Specific changes made in response to the current round of feedback are detailed in blue text below each corresponding point. We include line numbers corresponding to the second-round revised manuscript, which shows only the new tracked changes made since the previous submission. Screenshots are included to locate the relevant changes.

Thank you for these additional recommendations.

On behalf of all authors,
Grace Skinner

Reviewer #2

Remarks to the Author

The responses to both of the reviewers are thorough and I do not see any further issues in the revised manuscript. I also ran some of the new code and did not encounter any problems.

We thank Reviewer #2 for their positive feedback and for confirming they have no further concerns. We also appreciate the valuable feedback they provided in the previous round.

Reviewer #3

Remarks to the Author

This meta-analysis is a great contribution to our understanding of insect declines! It is well performed and documented, and the authors seem to have properly addressed most previous referees concerns. I believe the statistical analysis is sound, and the authors have properly considered potential bias and non-independence. I have just a couple of comments.

Thank you for taking the time to review our manuscript. We appreciate the encouraging and positive comments, and we have addressed the comments you have raised below.

First, I am concerned about their restriction of the taxa included. I understand that they were focusing on taxa more likely to be affected by invasive species based on an unpublished manuscript. But I cannot access the manuscript and I cannot know how these taxa were assessed. Not being able to evaluate the manuscript used to exclude other taxa is relatively minor. A main concern for me is that they are excluding two very abundant and rich orders: Lepidoptera and Diptera. These two orders together can account for about 300,000 species, whereas Orthoptera and Hemiptera together account for about 100,000 species. Thus, because they are not including almost a third of the insects, I believe the authors cannot say this is a test of the effect of invasive species on terrestrial insects. And they also did not find a negative effect on Coleoptera, the most diverse insect order. Lastly, by focusing on orders more likely to be affected by invasive species, they might be more prone to finding negative effects. This restriction could be by itself a bias in their study. I suggest toning down some of the main conclusions (e.g., line 332 and title).

Thank you for raising this point regarding the taxonomic scope of the meta-analysis. We clarify that our results are not necessarily representative of *all* terrestrial insects, but rather the subset included in our study. We now explicitly state this in the abstract (L32), at the start of the last paragraph of the

Introduction (L83), and in both the first sentence (L200-203) and the concluding paragraph of the Discussion (L344).

32 effects. We show that invasive alien species reduce the abundance of insects included in our study
33 abundance by 31%, and species richness by 26%, though these impacts are highly variable across

33 Here we present a meta-analysis of the impact of invasive alien species on a subset of terrestrial
34 insect biodiversity. While previous research has examined the effect of invasive alien species

Discussion

Here we show that, for the subset of terrestrial insect orders included in our study (Hymenoptera,
Coleoptera, Orthoptera, and Hemiptera), invasive alien species reduce terrestrial insect
(Hymenoptera, Coleoptera, Orthoptera, and Hemiptera) abundance by 31%; and terrestrial insect
species richness by 26%. However, the results are highly variable and context-dependent,

Here we provide clear evidence that invasive alien species have overall negative, yet highly
variable, effects on the abundance and species richness of terrestrial insects included in our study.

Regarding the absence of a negative effect for Coleoptera, this is reflected in the 'highly variable impacts' highlighted in our manuscript's title. To reinforce this narrative, we now emphasise this variability in the Abstract (L33) and the Discussion (L203 and L343-344).

32 effects. We show that invasive alien species reduce the abundance of insects included in our study
33 abundance by 31%, and species richness by 26%, though these impacts are highly variable across
34 taxa. with sStronger negative impacts were found from invasive alien animals compared to

203 species richness by 26%. However, the results are highly variable and context-dependent,
204 consistent with previous meta-analyses³⁵⁻³⁷. Although tests indicate some publication bias in the

343 Here we provide clear evidence that invasive alien species have overall negative, yet highly
344 variable, effects on the abundance and species richness of terrestrial insects included in our study.

We recognise that focusing on expert-selected groups may bias results toward detecting negative impacts. We expand the Discussion to highlight that broader inclusion of terrestrial insects might reveal a more variable and on average less negative response (L208-210). Given it has now been published, we now also cite and paraphrase our recent paper in *Diversity and Distributions* (citation 43 in our manuscript), where we explain some of our rationale and approach for the expert ranking of threats (Methods L359, Introduction L89). Wherever we have previously cited the in-preparation expert elicitation paper, we now instead cite this paper in *Diversity and Distributions*".

208 We note that broader inclusion of terrestrial insect orders beyond those identified as having
209 invasive alien species ranked among their top threats might reveal a more variable and on average
210 less negative response. The most substantive impacts of invasive alien species across these insect

alien species present (Fig. 5). We focused on primarily terrestrial insect orders for which invasive
alien species had previously been identified by experts as a major potential threat⁴³ (~~Bladon et al.~~
~~[manuscript in preparation]~~). This assessment evaluated 12 insect orders (Lepidoptera,

87 and wasps), and Orthoptera (grasshoppers, locusts, and crickets). We selected these orders
88 because invasive alien species were identified as a major potential threat in an expert elicitation
89 process⁴³ (~~Bladon et al. [manuscript in preparation]~~). We address two key research questions: 1.

Second, I believe this meta-analysis is novel in focusing solely on insects and including all types of invasive species, but there are other meta-analyses that have evaluated the effect of invasive species on insects. I found at least four meta-analyses that consider the impact of invasives on animals (not just insects, but at least a good portion of the taxa evaluated are insects or they have subdivisions by taxon), and one specific about Lepidoptera. These meta-analyses should be considered to be included in the paragraph starting in line 56 (and maybe might be useful for the rest of the manuscript as well):

Charlebois, J. A., & Sargent, R. D. (2017). No consistent pollinator-mediated impacts of alien plants on natives. *Ecology Letters*, 20(11), 1479-1490.

Fletcher, R. A., Brooks, R. K., Lakoba, V. T., Sharma, G., Heminger, A. R., Dickinson, C. C., & Barney, J. N. (2019). Invasive plants negatively impact native, but not exotic, animals. *Global Change Biology*, 25(11), 3694-3705.

Yoon, S. A., & Read, Q. (2016). Consequences of exotic host use: impacts on Lepidoptera and a test of the ecological trap hypothesis. *Oecologia*, 181(4), 985-996.

Montero-Castaño, A., & Vilà, M. (2012). Impact of landscape alteration and invasions on pollinators: a meta-analysis. *Journal of Ecology*, 100(4), 884-893.

Schirmel, J., Bundschuh, M., Entling, M. H., Kowarik, I., & Buchholz, S. (2016). Impacts of invasive plants on resident animals across ecosystems, taxa, and feeding types: a global assessment. *Global change biology*, 22(2), 594-603.

Thank you for highlighting these relevant meta-analyses. We agree that these studies provide valuable context and further support for our findings of negative but highly variable impacts of invasive species on insects. We now cite three of these studies (Fletcher et al., 2019; Montero-Castaño and Vilà, 2012; Schirmel et al., 2026) where you suggest in the Introduction (L60-62), and again in the Discussion (L203-204) to support our conclusion of 'negative but highly variable' effects. We also cite Yoon and Read (2016) in the final Introduction paragraph (L84-85). We chose not to cite Charlebois and Sargent (2017) as their focus is primarily on how invasive plants affect native plants via pollinator-mediated mechanisms, rather than directly on insect pollinators themselves.

50 invasive alien species on insects specifically³¹⁻³⁴. ~~Several other meta-analyses have considered the~~
51 ~~impact of invasive alien species on animals more broadly—including, but not focusing on,~~
52 ~~insects—often highlighting negative but highly variable effects~~³⁵⁻³⁷. Syntheses have also

203 species richness by 26%. However, the results are highly variable and context-dependent,
204 consistent with previous meta-analyses³⁵⁻³⁷. Although tests indicate some publication bias in the
34 insect biodiversity. While previous research has examined the effect of invasive alien species
35 (specifically invasive alien plants) on Lepidoptera⁴⁹, ~~W~~e focus on insects in the primarily
86 terrestrial orders Coleoptera (beetles), Hemiptera (true bugs), Hymenoptera (ants, bees, sawflies,
87 and wasps), and Orthoptera (grasshoppers, locusts, and crickets). We selected these orders

Remarks on code availability

Code seems okay.

We appreciate the reviewer's assessment of the code. No changes were made to the code during this revision.